# Insulin-Degrading Enzyme Is a Non Proteasomal Target of Carfilzomib and Affects the 20S Proteasome Inhibition by the Drug

**DOI:** 10.3390/biom12020315

**Published:** 2022-02-16

**Authors:** Grazia Raffaella Tundo, Diego Sbardella, Francesco Oddone, Giuseppe Grasso, Stefano Marini, Maria Grazia Atzori, Anna Maria Santoro, Danilo Milardi, Francesco Bellia, Gabriele Macari, Grazia Graziani, Fabio Polticelli, Paolo Cascio, Mariacristina Parravano, Massimo Coletta

**Affiliations:** 1IRCCS-Fondazione BIETTI, 00198 Rome, Italy; francesco.oddone@fondazionebietti.it (F.O.); mariacristina.parravano@fondazionebietti.it (M.P.); 2Department of Chemical Science, University of Catania, 95125 Catania, Italy; grassog@unict.it; 3Department of Clinical Sciences and Translational Medicine, University of Roma Tor Vergata, 00133 Rome, Italy; stefano.marini@uniroma2.it; 4Department of Systems Medicine, University of Rome Tor Vergata, 00133 Rome, Italy; mariagraziaatzori.2@gmail.com (M.G.A.); graziani@uniroma2.it (G.G.); 5Institute of Crystallography, CNR, 95126 Catania, Italy; amsantoro@unict.it (A.M.S.); danilo.milardi@cnr.it (D.M.); francesco.bellia@cnr.it (F.B.); 6Department of Sciences, Rome Tre University, 00146 Rome, Italy; gabriele.macari@uniroma3.it (G.M.); polticel@uniroma3.it (F.P.); 7National Institute of Nuclear Physics, Rome Tre Section, 00146 Rome, Italy; 8Department of Veterinary Sciences, University of Turin, 10095 Grugliasco, Italy; paolo.cascio@unito.it

**Keywords:** proteasome, carfilzomib, insulin-degrading enzyme, cancer, neurodegeneration

## Abstract

Carfilzomib is a last generation proteasome inhibitor (PI) with proven clinical efficacy in the treatment of relapsed/refractory multiple myeloma. This drug is considered to be extremely specific in inhibiting the chymotrypsin-like activity of the 20S proteasome, encoded by the β5 subunit, overcoming some bortezomib limitations, the first PI approved for multiple myeloma therapy which is however burdened by a significant toxicity profile, due also to its off-target effects. Here, molecular approaches coupled with molecular docking studies have been used to unveil that the Insulin-Degrading Enzyme, a ubiquitous and highly conserved Zn^2+^ peptidase, often found to associate with proteasome in cell-based models, is targeted by carfilzomib in vitro. The drug behaves as a modulator of IDE activity, displaying an inhibitory effect over 10-fold lower than for the 20S. Notably, the interaction of IDE with the 20S enhances in vitro the inhibitory power of carfilzomib on proteasome, so that the IDE-20S complex is an even better target of carfilzomib than the 20S alone. Furthermore, *IDE* gene silencing after delivery of antisense oligonucleotides (siRNA) significantly reduced carfilzomib cytotoxicity in rMC1 cells, a validated model of Muller glia, suggesting that, in cells, the inhibitory activity of this drug on cell proliferation is somewhat linked to IDE and, possibly, also to its interaction with proteasome.

## 1. Introduction

The proteasome is the central proteolytic assembly of the Ubiquitin Proteasome System (UPS), a main catabolic pathway which carries out the targeted proteolysis of the majority of intracellular proteins, mostly short-lived, being thus involved in the regulation of all cellular metabolic processes [1,2]. The hierarchical organization of the UPS encompasses two intertwined steps, namely (i) the covalent attachment of ubiquitin moieties to a given substrate through a three-step reaction catalyzed by E1-E2-E3 enzymes, and (ii) its degradation by the proteasome [2,3,4]. The holoenzyme is composed of the 20S core particle (CP), a barrel-shaped particle made up by (a) two outer α-rings (each hosting the α1-α7 subunits), which allosterically regulate the open/closed configuration of the gate through which substrates are grasped and pulled for catalysis, and (b) two inner β rings (each hosting β1-β7 subunits) [3,4,5], where β1, β2 and β5 subunits hold the catalytic sites for the caspase-like, trypsin-like and chymotrypsin-like enzymatic activities, respectively [2].

Proteasome exists in vivo in different structural configurations, referred to as capped assemblies, wherefore a 20S particle is capped at one or both of the α-ends by the 19S regulatory particles (RP). The 19S couples the ATP-hydrolysis with the binding, unfolding and translocation of the substrate (e.g., poly-ubiquitinated proteins) into the 20S catalytic core. Over recent decades, regulatory particles other than the 19S have been described, namely PA28 protein family and Blm10/PA200, which confer to the 20S with different but interconnected substrate specificities and biological roles that are not completely understood yet [6,7,8,9,10,11].

Proteasome pathway is extremely dynamic and heterogeneous, since proteasome composition, specificity and activity are adaptable to cellular metabolic needs and, thus, finely regulated at multiple steps, including post-translational modifications and regulatory factors, such as Proteasome Interacting Proteins (PIPs) [1,4,9,10,11]. In this regard, it has been reported that Insulin-Degrading Enzyme (IDE), a Zn^2+^-dependent peptidase highly conserved through evolution and ubiquitously expressed in human tissues, interacts and modulates the activity of the 20S, potentially competing with the 19S, at least in vitro [12,13,14,15]. Historically, IDE’s biological role has been associated to insulin turnover and much evidence has suggested that IDE dysfunction should lead to age-dependent glucose intolerance and to the onset of the type 2 diabetes (T2DM) [16,17]. In this framework, IDE has been long considered a promising therapeutic target to slow down insulin catabolism, limiting the progression of diabetes and related complications, such as diabetic retinopathy [17,18,19]. Besides insulin, IDE is involved in the degradation of amyloidogenic proteins (e.g., β-amyloid), providing evidence of its involvement in Alzheimer’s disease (AD) pathogenesis [20,21,22]. Furthermore, IDE contribution to AD fits well with the novel biological roles that have been continuously described over the last decade, which encompass, namely: (i) the catalytic activity on amyloidogenic substrates, (ii) the “dead-end chaperone” hypothesis and (iii) modulation of the UPS [13,14,23,24,25].

Within the proteostasis network, it is well known that dysregulation of the proteasome activity destabilises and/or disrupts the balance between tumour suppressors and oncoproteins, promoting carcinogenesis and cancer progression [1,26]. Therefore, in accordance with the “proteotoxic-crisis” hypothesis, proteasome has gained increasing interest as an efficient target for anti-neoplastic drugs. As a matter of fact, three clinically approved proteasome inhibitors (PIs) are currently available, namely (a) bortezomib (BTZ) (Velcade), (b) carfilzomib (CFZ) (Kyprolis), and (c) the first oral PI, ixazomib (IXZ) (Ninlaro). “First generation PI” BTZ is a reversible inhibitor of chymotrypsin-like activity of the 20S and its introduction has revolutionised the approach to multiple myeloma (MM) treatment, leading to improvement of patients’ survival rate. However, its therapeutic activity is limited by several molecular and clinical drawbacks, including drug resistance development, occurrence of side effects, low potency and specificity; in fact, it retains activity also toward non-proteasome targets [27,28,29,30,31].

Conversely, CFZ seems to overcome these limitations by virtue of its chemical composition, since it is a tetrapeptide with a terminal epoxyketone group, scaffolded on the natural compound epoxomicin, thought to be the most specific proteasome inhibitor. CFZ displays an inhibitory power equivalent to that of BTZ for chymotrypsin-like subunits (IC50 = 6 nM), whereas caspase-like and trypsin sites are only very weakly inhibited [32,33]. The dramatically enhanced specificity of CFZ for proteasome is due to the fact that it forms a covalent adduct between its C-terminal ketone moiety and the Thr1O residue of each inhibited subunit of 20S and, unlike peptide boronates (such as BTZ and IXZ), it forms a stable morpholine ring between Thr1 N-terminal amino group and epoxide α carbon. This high selectivity was supposed to guarantee a higher tolerability and a more favourable pharmacological profile than BTZ. However, several patients display or develop CFZ resistance and suffer from a number of adverse events only partially shared with BTZ [1,34,35]. One of the main reasons of the different toxicity profile should be searched for by looking into the different non-proteasome targets of these two drugs. As a matter of fact, a recent work correlates CFZ-induced cardiotoxicity with downregulation of autophagy pathway [36]. However, a specific non-proteasome target of CFZ has not been reported yet.

In this work, by combining biochemical, molecular and bioinformatic tools, we report that CFZ is an IDE ligand in vitro, exerting some inhibition of catalytic activity on synthetic and natural (insulin) substrates. Furthermore, we show that CFZ’s inhibitory power in vitro is influenced by IDE interaction with the 20S. Finally, IDE gene silencing after delivery of a pool of antisense oligonucleotides (siRNA), significantly lowers CFZ cytotoxicity in rMC1 cells, a validated rat cellular model of Muller glia.

## 2. Materials and Methods

### 2.1. Characterisation of CFZ Effect on the Chymotryptic-like Activity of 20S in the Absence and in the Presence of IDE

1 nM 20S proteasome (Boston Biochem, Cambridge, MA, USA) was incubated with increasing concentrations of CFZ (ranging from 5 to 60 nM) for 10 min at 37 °C in the assay buffer (25 mM Tris-HCl, pH 7.6). Thereafter, 50 µM 7-amino-4-methylcoumarin (AMC) labeled Suc—Leu—Leu—Val—Tyr—AMC peptide (referred to as LLVY-amc) (Boston Biochem, Cambridge, MA, USA) was added. Proteasome activity was monitored by measuring the AMC fluorescence at 460 nm (excitation at 380 nm) over 60 min, until linearity was observed. Data are expressed as normalised percentages of residual activity considering the slope of the control (fluorogenic peptide/proteasome in the absence of CFZ) as 100% of proteasome activity. Dose-response plots of the residual proteasome activity in the presence of increasing concentrations of the inhibitor provide a quantitative estimate of its potency.

For data in the presence of 20 nM IDE (Merck-Millipore, Burlington, MA, USA), IDE and 20S proteasome (1 nM) were incubated for 10 min at 37 °C in the assay buffer; thereafter, the desired amount of CFZ was added and the reaction mixture was incubated for an additional 10 min at 37 °C. Finally, the fluorogenic substrate was added and the residual activity measured.

The purity of 20S and IDE preparations was assayed by SDS-PAGE under denaturing and reducing conditions before performing all reported experiments (see Appendix A).

### 2.2. Kinetics Analysis of CFZ Effect on IDE

A total of 10 nM of IDE was incubated in the presence or in the absence of increasing concentrations of CFZ in 25 mM Tris-HCl, pH 7.6 at 37 °C for 15 min. The characterization of IDE enzymatic activity was carried out employing a fluorogenic substrate specific for its activity (i.e., MCA-RPPGFSAFK(Dnp)-OH) (Merck-Millipore, Burlington, MA, USA). The rate of hydrolysis of the fluorogenic substrate (excitation and emission wavelengths at 320 and 405 nm, respectively) was monitored over 1 h in a fluorimeter (Cary-Eclipse, Varian, Santa Clara, CA, USA) and the velocities were calculated over the linear range of peptide hydrolysis (typically within the first 10 min of reaction). Over the same interval, no auto-proteolysis of the substrate was observed.

Data are expressed as normalised percentages of residual activity considering the slope of the control (fluorogenic peptide/IDE in the absence of CFZ) as 100% of IDE activity. Dose-response plots of the residual IDE activity in the presence of increasing concentrations of the inhibitor provide a quantitative estimate of its potency.

### 2.3. Molecular Docking

Three sets of molecular docking simulations were performed using Autodock Vina [37]. The initial structure and conformation of CFZ was obtained, using SMILES representation, through the opensource toolkit for cheminformatics RDKit by using the ETKDG method (https://rdkit.org, accessed on 1 December 2021). The IDE three-dimensional structure was retrieved from the Protein Data Bank (PDB code 2JG4). The protein and the ligand were prepared for molecular docking (e.g., merging charges, adding polar hydrogen, etc.) using the AutoDockTools’ scripts prepare_receptor4.py and prepare_ligand4.py, respectively. The first set of molecular docking simulations was performed blindly, that is extending the search grid to the whole protein. After this step, the second set of simulations was performed using a receptor grid of 10 Å × 10 Å × 10 Å centered around the binding region identified in the previous set of simulations (i.e., the zinc binding site, see Results). The last set of simulations was performed with the same grid setting as the previous one but allowing some flexibility to residues surrounding the zinc ion in the inner cavity of the protein, namely Phe820, Arg824, Asn139, Phe141, Thr142, Ser143, Ser96, Trp199, Thr220, and Asn193.

### 2.4. Mass Spectrometry

The effect of CFZ on the IDE-mediated hydrolysis of insulin has been investigated by applying an experimental approach based on the LC-MS characterisation of insulin hydrolytic peptides [38]. Briefly, IDE (100 nM) was incubated in the presence or in the absence of increasing concentrations of CFZ (from 40 nM to 10 µM) in phosphate buffer 10 mM (pH 7.4) for 15 min at 37 °C. Insulin (5 µM) was added and samples were collected for 90 min; the reaction was quenched with TFA 0.5%. The peptides content was analysed using a hybrid quadrupole-Orbitrap mass spectrometer (Q-Exactive, Thermo Scientific) coupled to the Ultimate 3000 HPLC RSLCnano system (Dionex, Thermo Scientific) through the EASY-Spray source (Thermo Scientific) by using the instrument settings previously described [39].

### 2.5. Cell Culture and Gene Silencing

Rat Muller glia cells (rMC1) were routinely cultured in low-Glucose (5 mM) DMEM supplemented with 10% FBS and antibiotics under standard aerated conditions (37 °C, 5% CO_2_).

For gene silencing studies, cells were seeded at a density of 6 × 10^3^ cells/well in a 96-well Transwell cell plate (Corning, NY, USA) in standard medium.

The day after seeding, the cell monolayer was washed once with pre-warmed 1x PBS and the reduced-serum medium, referred to as OptiMEM (Thermofisher Scientific, MA, USA) was added, following the manufacturer’s instructions. This medium was supplemented with 50 nM small-interfering RNA (siRNA oligonucleotides, Dharmacon, Thermofisher Scientific, MA, USA), or the vehicles they were dissolved in (referred to as control). A pool of 4 oligonucleotides targeting rat IDE and non-targeting sequences were used. After 48 h from siRNA delivery, CFZ was added at different concentrations as indicated in figures. DMSO (CFZ vehicle) was used as internal control.

After additional 24 h (i.e., 72 h from siRNA delivery), MTS tetrazolium assay (Sigma Aldrich, St. Louis, MO, USA) was used to test the cytotoxicity of CFZ under the different experimental conditions.

### 2.6. Native Gel Electrophoresis and Western Blotting

Crude cell extracts (e.g., soluble fraction of the cell) were isolated from cells treated with CFZ and DMSO (CFZ vehicle) under non-denaturing condition through freeze-thawing cycles in 250 mM sucrose, 20% glycerol, 25 mM Tris-HCl, 5 mM MgCl_2_, 1 mM EDTA, 1 mM DTT, 2 mM ATP, pH 7.4 supplemented with protease inhibitor cocktail [40]. Thereafter, lysates were cleared by centrifugation at 13,000 rpm, for 20 min at 4 °C and protein concentration was normalized by Bradford assay. For each experimental condition, 75 µg of proteins were separated under native conditions into a 3.5% acrylamide gel. Gels were then harvested and soaked in a clean dish in the reaction buffer (50 mM Tris, 5 mM MgCl_2_, 1 mM ATP, pH 7.5) supplemented with 100 μM LLVY-amc. Proteins were then transferred to a HyBond-ECL nitrocellulose filters and probed with antibodies specific for 20S proteasome subunit α7, 19S proteasome subunit Rpt5 and IDE (Protein-tech Group, Manchester, UK), diluted 1:3000 in 0.02% Tween-PBS fat-free milk; afterwards, they were incubated with a Horseradish Peroxidase-conjugated anti-rabbit or anti-mouse IgG antibody (Biorad, Hercules, CA, USA), diluted 1:50,000 in 0.2% Tween-PBS fat-free milk.

For denaturing and reducing WB, cell pellets were lysed in RIPA buffer supplemented with protease inhibitor cocktail and cleared by centrifugation at 13,000 rpm for 30 min at 4 °C. Protein concentration was normalized by Bradford assay. For each lane, a minimum of 15 µg of total proteins were loaded. Protein transfer to filters was carried out as described in the previous paragraph. Then, filters were probed with antibodies specific for IDE and ubiquitin (Protein-tech Group, Manchester, UK), diluted 1:3000 in 0.02% Tween-PBS fat-free milk and developed as described in the previous section. Protein bands were detected by ECL and acquired through an Azure-Biosystem, C280, gel documentation system. β-actin was used as internal control after having verified that protein detection did fall within the linear dynamic range.

### 2.7. Statistical Analysis

The figures illustrate a representative experiment of a minimum of three independent analysis and data are reported as Mean ± SD unless otherwise indicated. Statistical analysis was performed by One-way ANOVA followed by Tukey’s post-hoc significance test.

## 3. Results

### 3.1. Effect of CFZ on 20S and on IDE-20S Complex Activity In Vitro

CFZ is well known to be an efficient and specific inhibitor of 20S proteasome activity [41,42], as it clearly emerges from the dependence of the 20S chymotrypsin-like activity on CFZ concentration (Figure 1). However, a non-linear least-squares fitting of data considering CFZ as a simple competitive inhibitor is clearly unsatisfactory (see dashed line “a” in Figure 1). Thus, the dependence of residual activity on CFZ concentration is definitely steeper than what expected for a simple inhibitor-binding process. The simplest explanation for this behaviour is to take into account that each 20S molecule has two chymotrypsin-like active sites; therefore, two CFZ molecules bind to one 20S molecule, and CFZ binding to 20S can be analyzed according to the following Figure 1:

The analysis of data in Figure 1 has been carried out employing the following Equation (1):(1)$$\mathrm{Residual}\;\mathrm{Activity}=\frac{K_{I}^{2}\cdot \delta +K_{I}\cdot \delta \cdot \left[CFZ\right]}{K_{I}^{2}\cdot \delta +2\cdot K_{I}\cdot \delta \cdot \left[CFZ\right]+{\left[CFZ\right]}^{2}}$$

From Figure 1 it is evident that *δ* = 1 (dashed curve *a*) is not suited to describe data and that *δ* << 1 is required (see Table 1). This clearly indicates that there is a functional relationship between the two chymotrypsin-like catalytic sites, such that CFZ binding to either one of the two sites brings about a conformational change, rendering the partner site more prone to CFZ binding.

As already reported, in the presence of 20 nM IDE the activity of the 20S is partially inhibited [13]; starting from this inhibition degree, the addition of CFZ further inhibits the chymotrypsin-like activity of the 20S (Figure 1). However, the dependence on CFZ concentration (continuous line) is different from that expected in the case of no effect by IDE binding (see dashed line “b”). In particular, in the IDE-bound 20S, the affinity of CFZ for the first catalytic site is somewhat lower with a decrease of free energy change by 1.2 kJ/mol (see Table 1), but the cooperative interaction between the two catalytic sites of 20S is enhanced in the presence of IDE, giving rise to a larger interaction energy by over 3 kJ/mol (see Table 1). Therefore, IDE binding to the 20S likely induces a structural change, which on one side slightly reduces the ligand affinity for the first site, but on the other side enhances the positive interaction energy between the two catalytic sites: this interpretation is also perfectly in accordance with the observed effect of IDE on the 20S enzymatic activity, as reported elsewhere [13]. As a whole, in the presence of IDE, CFZ turns out to be a more powerful inhibitor of 20S proteasome.

### 3.2. Molecular Docking of the IDE-CFZ Interaction

Since the presence of CFZ alters IDE–20S interaction in vitro, we have tested the effect of CFZ on IDE through a combined approach of biochemical (see the next sections) and molecular docking studies. The interaction of CFZ with IDE and its molecular determinants were investigated first through a set of blind molecular docking simulations, that is, without making any assumption about the binding site of the compound. Interestingly, CFZ was predicted to bind in the inner cavity of the protein, near the zinc ion-binding site. To refine these results, a second round of simulations was carried out, centering the search grid on the zinc ion (see Methods for details). The predicted binding affinity value for the best pose observed for CFZ was −8.9 kcal/mol (Table 2). Finally, to refine the prediction and better investigate the binding mode of the ligand, a set of protein residues surrounding the binding region, identified in the previous simulations, were treated as flexible and novel simulations were performed. The best docking poses obtained in this last step (Figure 2) displayed an increase in the predicted binding affinity values up to −9.9 kcal/mol for CFZ.

### 3.3. CFZ Modulates IDE Activity

Since MD studies reveal that CFZ can bind IDE, we have investigated the effect of CFZ on IDE activity using a specific fluorogenic substrate.

A first important result is the evidence that CFZ is not a substrate for IDE, since incubation of CFZ with IDE for several hrs did not produce any fragmentation of CFZ, as from mass spectrometry analysis (see Appendix A).

The dependence of the rate of IDE enzymatic activity on the CFZ concentration is shown in Figure 3 at different substrate concentrations. In Figure 3, the relative velocity corresponds to the ratio between the enzymatic rate at a given CFZ concentration (i.e., ^CFZ^(*v*/[*E*_0_])) with respect to the rate in the absence of CFZ (i.e., ^0^(*v*/[*E*_0_]) for the same substrate concentration. A first observation on data, reported in Figure 3, is that the CFZ concentration over which an effect on IDE is observed is much higher than that reported for the 20S (Figure 1). Therefore, we can reasonably assume that there is no CFZ binding to IDE, under the conditions described in Figure 1, and that the IDE-linked effect on 20S inhibition by CFZ is only referable to the interaction of IDE with the 20S and to the corresponding conformational change [13]. Focusing on CFZ-IDE interaction, we observe that CFZ addition brings about an enhancement of the substrate hydrolysis rate for [CFZ] ≤ 0.3 µM, followed by an inhibitory effect when CFZ concentration rises up. However, this activation is observed only for [substrate] ≤ 10 µM, while at higher substrate concentrations, the inhibitory behavior is predominant at each CFZ concentration (Figure 3). Each experimental point in Figure 3 corresponds to the average result of (at least) four independent experiments, and bars refer to the observed variation of the effect, clearly demonstrating its statistical relevance. From the phenomenological viewpoint, this effect envisages the existence of (at least) two binding sites for CFZ on IDE, one of which activates its enzymatic activity (likely through an allosteric mechanism) and a second one (probably the active site of IDE) whose occupancy by CFZ brings about a competitive inhibition of the enzymatic activity. This mechanism may appear in contrast with the results of Molecular Docking (see above), which suggest that CFZ preferentially binds to IDE catalytic site without interacting with the exosite. However, this allosteric activating effect at [CFZ] ≤ 0.3 µM suggests the possibility that the highest affinity CFZ binding site to IDE exerts in solution some allosteric effect on the assembly of IDE [43], inducing a more active conformation, a mechanism which cannot be detected by molecular docking. The occurrence of a second site can be phenomenologically described according to the following Figure 2:

On the basis of this mechanism, we can give a more detailed explanation for the substrate-dependent (i.e., on [S]) CFZ-linked effect on the relative velocity (see Figure 3), since the values of ^0^*k*_cat_ and ^0^*K*_m_ are already known [44] and from data, reported in Figure 3, it is possible to obtain all other parameters to characterize (Equation (2)), which are reported in Table 3.
(2)Relative Velocity=(v[E0])CFZ(v[E0])0=(K0m⋅KCFZ2+KCFZ2⋅[S])⋅(KCFZ1⋅α+β⋅[CFZ])K0m⋅α⋅(KCFZ1⋅KCFZ2+KCFZ2⋅[CFZ]+[CFZ]2)+KCFZ2⋅[S]⋅(KCFZ1⋅α+[CFZ])

Therefore, from data reported in Figure 3, it looks like that CFZ is not primarily a competitive inhibitor of IDE, since its main effect at low CFZ concentrations induces the activation of IDE. It is only at higher concentrations that CFZ behaves as a competitive inhibitor, occupying the catalytic site of IDE and interfering with substrate bonding and processing.

### 3.4. Mass Spectrometry Analysis of Insulin Degradation by IDE in the Presence of CFZ

IDE is the main enzyme involved in the insulin clearance, and alteration of its activity is associated to TDM2 onset and progression [19,45]. Therefore, we have tested whether CFZ modulates IDE activity also toward a macromolecular substrate, such as insulin. IDE was incubated in the presence or in the absence of indicated CFZ concentrations in phosphate buffer 10 mM, pH 7.4, for 15 min at 37 °C, and then 5 μM of freshly prepared insulin was added. Insulin degradation was followed over 90 min and aliquots were collected and analyzed at indicated time intervals. The rate of formation of two main insulin fragments, B1-9-A1-13 and B1-30/A12-21, produced by the proteolytic activity of IDE on whole insulin, was monitored. As comes out from Figure 4, CFZ induces a very small but statistically significant (about 10%) concentration-dependent decrease in the rate of formation of these two fragments. It is important to highlight that, besides the differences in the substrate used (insulin in this case), the experimental conditions of this experiment are quite different from the ones used in the assay with the fluorogenic substrate. Indeed, in the MS assays with insulin, IDE concentration was 10-fold higher (100 nM) in order to achieve the needed insulin degradation rate detectable by MS within the assay time; therefore, the oligomeric IDE distribution is very likely to be altered, bringing about some variation in the CFZ interaction. Since the oligomeric state of IDE plays a major role on IDE activity, it is not straightforward to compare the results of the two sets of data obtained with the fluorogenic substrates and with full-length insulin. However, our results indicate that CFZ slightly inhibits IDE activity also in the case of insulin. Unlike what reported for the fluorogenic peptide, no activating effect of CFZ is observed, at least under these experimental conditions.

Notably, the concentrations of the hydrolytic fragments, produced by IDE and monitored by the intensity of their peak areas in the mass spectra, increase with time during insulin incubation with IDE, as expected. However, long insulin fragments (such as the B1-30/A12-21) are also IDE substrates themselves and can be catalytically degraded by the enzyme soon after they are formed. For this reason, the intensity of their peaks depends not only on the rate of their formation (from the degradation of whole insulin by IDE), but also on the rate of their degradation into shorter peptides (secondary cleavage sites) [46]. The abundancy of these longer fragments in the solution will therefore reach a maximum during the time course experiments and will decrease at longer incubation times (in our experimental conditions this situation already occurs for incubation times as long as 90 min).

### 3.5. Effect of CFZ on IDE-20S Interaction in rMC1 Cells

In order to find out whether the molecular interaction between IDE and 20S (see Figure 1) and its effect on CFZ inhibition has a physiological relevance, we investigated whether CFZ cytotoxicity is affected by IDE intracellular content in living cells. To this aim, rMC1 (rat Muller glia cell line) was selected for its recognized role in the pathogenesis of glaucoma and diabetes complications, such as neuropathy and retinopathy that are research topics inherent to IDE biology [14,43,44,45]. First, the 50% inhibitory concentration (IC_50_, which is defined as the drug concentration that reduced the cell viability by 50% when compared to untreated controls) of CFZ on rMC1 cells was determined (Appendix A). This analysis highlighted some intrinsic resistance of rMC1 to CFZ: the calculated IC_50_ was around 453.9 nM, which is fairly higher than that observed for MM cells but consistent with that reported for some adherent cell lines in vitro [47].

Thereafter, the occurrence of IDE interaction with proteasome particles in resting rMC1 and in cells exposed to 150 nM and 500 nM CFZ (representing a sub-lethal and a lethal concentration of the drug, respectively) was assayed by native-gel electrophoresis coupled with WB over a brief time interval (1 h and 2 h) during which apoptosis-related pathways induced by drug exposure are unlikely to introduce unpredictable off-target effects. The activity of proteasome assemblies populating the crude cell extracts was probed in-gel with LLVY-amc synthetic substrate (Figure 5A). As expected, CFZ abolished proteasome activity on the synthetic substrate under all tested conditions. However, in this cellular model, immunostaining of proteasome particles cast light on two unexpected findings:(i)In the presence of CFZ, proteasome immunostaining was markedly reduced (Figure 5B); this effect was observed both for free 20S (α7 immunostaining) and for the capped assemblies and free 19S (Rpt5 immunostaining). To rule out uneven gel loading, filters were stained with Ponceau S after transfer and α7 subunit was analyzed by denaturing and reducing WB, revealing a pattern consistent within all the lanes (Figure 5C);(ii)IDE distribution across the mass/charge range of complexes was significantly altered by CFZ (Figure 5D). In particular, immunostaining of IDE dimer (red arrow) significantly dropped in the presence of CFZ with the exception of cells treated with 150 nM CFZ for 1 h. Under all experimental conditions tested, a progressive increase of IDE co-localization with the single-capped species was observed (Figure 5D). This accumulation peaked up in the presence of 500 nM CFZ for 2 h. Nevertheless, unlike many other cell lines tested so far [13], rMC1 cells were characterized by a robust IDE immunostaining in correspondence of the single capped proteasome, but not of free 20S. To rule out that accumulation of IDE on CFZ-inhibited proteasome was a phenomenon shared by other putative PIPs, filters were further probed with an anti-HSP70 antibody. Immunostaining of this chaperone, which was well distributed across different proteasome assemblies, was unaffected by CFZ treatment (Figure 5E).

As expected, IDE immunostaining under denaturing and reducing conditions was fully comparable under all tested experimental conditions (Figure 5F), and poly-ubiquitinated proteins accumulated with the exception of cells treated with 150 nM CFZ for 1 h (Figure 5G).

These features indeed suggest that (i) proteasome inhibition by CFZ likely leads to a faster degradation of all proteasome populations, and (ii) interaction with CFZ brings about an enhancement of IDE affinity for single capped proteasome particles, likely coupled to the CFZ-linked conformational change (see Figure 1).

Therefore, rMC1 cells were preliminary silenced for IDE expression by delivery of anti-sense oligonucleotides (siRNA) at 50, 75 and 100 nM final concentration. As internal controls, rMC1 cells were challenged with the siRNA vehicle (generally referred to as Control, as untreated and vehicle treated cells were preliminarily found to be fully comparable) and a pool of non-targeting siRNA (referred to as Pool) to rule out off-target effects. After 72 h from oligonucleotides delivery, cell lysates were harvested and probed for IDE and β-actin by WB. With respect to Control and Pool-treated cells, IDE immunostaining displayed a very robust drop in the presence of all siRNA concentrations tested (Appendix A). Thus, further assays were performed by delivering 50 nM siRNA.

Thereafter, since the culture medium used for silencing experiments is serum-reduced (referred to as Opti-MEM) and cells show limited proliferation with respect to cells cultured in standard medium, the IC_50_ of CFZ was calculated 72 h after medium replacement. In the serum-reduced medium, the CFZ IC_50_ was significantly lower than that previously observed under standard culture conditions, attesting to 228.7 nM (Appendix A). At this stage, we cannot clarify whether this significant drop of the IC_50_ is caused by the increased susceptibility of cells grown under low-serum conditions or reduced stability of CFZ in serum-rich conditions.

Thus, a tailored study was set up to verify the cytotoxicity of CFZ in the presence of IDE silencing.

To this aim, cells were challenged with 200 nM and 400 nM CFZ 48 h after siRNA delivery and cytotoxicity evaluated by MTS assay after 24 h (that is 72 h from siRNA delivery).

Unlike Control and Pool-treated cells, for which CFZ turned out to be robustly toxic under all tested concentrations, viability of IDE-silenced cells was not significantly compromised by delivery of 200 nM CFZ (Figure 5H). Conversely, delivery of 400 nM CFZ was associated with a significant toxicity also in IDE-silenced cells but at a significantly lower rate than Control or Pool-treated cells challenged with this higher CFZ concentration (Figure 5H).

In fact, the IC_50_ of CFZ turned out to be significantly higher in the case of IDE-silenced cells (326.1 nM) (Appendix A). Conversely, Pool-treated cells displayed an IC_50_ (237.7 nM) comparable to that of Control cells. These data clearly suggest that the presence of IDE interacting with proteasome (either 20S particles and/or capped particles) enhances the toxicity of CFZ, also confirming that in cells the IDE-linked increase of CFZ inhibitory power is operative (see Figure 1).

## 4. Discussion

The introduction of BTZ in cancer therapy for the treatment of patients affected by MM and other haematological malignancies has been a milestone in molecular medicine [1,28,31]. However, the applicability of this drug has been limited by several drawbacks, including the onset of resistance and the emergence of adverse events in treated patients [1,28,31]. These issues have been only partially overcome by the introduction of CFZ, a drug regarded as an extremely selective inhibitor of proteasome chymotrypsin-like activity, even though it has been proposed that an alkynyl analogue of CFZ has additional potential intracellular targets [42]. Furthermore, very recently, CFZ-induced cardiotoxicity has been associated to the inhibition of AMPKa/TORC1 pathways, but the detailed molecular mechanisms responsible for this effect have not been clarified yet [36]. In this paper, we report that CFZ modulates the activity of IDE, an enzyme originally discovered for its involvement in the degradation of insulin and β-amyloid and associated with TDM2 and AD pathogenesis [14,23]. However, IDE’s role in cell biology is wider than originally thought and several independent studies suggest its multifaceted contribution to the regulation of the proteostasis network through non-catalytic proteasome modulation and chaperone-like scavenger activities on aggregating-prone peptides [12,13,14,25,48]. With respect to proteasome, IDE was found to associate and modulate proteasome activity in vitro and in cell-based models, even though the patho-physiological significance of this association remains unclear [14,23].

Herein, we confirm that CFZ dramatically inhibits 20S proteasome activity in vitro, but it also emerges that its inhibitory activity is influenced by (i) the intramolecular functional relationship between the two active sites in the β5 subunits, and (ii) the interaction of 20S with IDE (Figure 1). Thus, in the presence of IDE, which is known to be also an inhibitor of proteasome enzymatic properties, at least at the investigated concentrations [13], CFZ turns out to be a better inhibitor of 20S, with a 1.2 kJ/mol decrease of free energy change for CFZ interaction (see Table 1 and Figure 1).

Concerning the interaction of CFZ with IDE, it is the first time that a last generation PI has resulted in being effective on IDE also, even though we must remark that features of this interaction display meaningful differences for IDE and 20S proteasome. Thus, from biochemical studies, employing a fluorogenic peptide, CFZ binds IDE at least in two sites (Figure 1). The binding to the first site activates IDE enzymatic properties, probably through an allosteric mechanism. Conversely, the binding to the second site, most likely the IDE active site, as it emerges from docking studies on IDE monomer (Figure 3), leads to a competitive inhibition of its enzymatic activity. It is important to recall that, in solution, IDE exists as a mixture of different forms, (i.e., monomer, dimer and tetramer), and that the dimer has been proposed to be the most active one [45]. Therefore, the first allosteric binding site of CFZ on IDE, which is not detected in this computational study, might be closely related to the structural assembly in solution and difficult to unveil in the deposited structures. In addition, this putative effect of CFZ on the IDE assembly seems somehow supported also by the evidence in the cellular model that the amount of dimeric IDE is markedly affected upon addition of CFZ (see Figure 5D).

On the basis of these considerations, CFZ looks more a modulator than a specific inhibitor of IDE enzymatic activity. Therefore, the very low effect of CFZ on insulin degradation by IDE (see Figure 4) may appear less puzzling. As a matter of fact, CFZ binding to the catalytic site for a competitive substrate inhibition is not the primary interaction and it might be displaced by insulin binding, which occurs through a very extended surface (involving both the catalytic site and the exosite, which is about 30 Å away from the active site). In this respect, the inhibitory effect, reported in Figure 3 for a small synthetic fluorogenic substrate, could be dramatically reduced for a macromolecular substrate, such as insulin, which displays a much higher affinity for IDE [49].

In this framework, the pharmacological studies on rMC1 may indirectly support the whole working hypothesis about a significant contribution of IDE to CFZ cytotoxicity. In fact, the very significant increase of CFZ IC_50_ in IDE-silenced cells, compared to that observed for Control and Pool-treated cells, envisage that drug cytotoxicity is somewhat related to IDE bioavailability.

At this stage, it cannot be unequivocally stated whether: (i) under normal growth condition and in the absence of IDE-silencing, CFZ cytotoxicity is determined by a synergism coupling inhibition of proteasome and inhibition/modulation of IDE, regardless of proteasome interaction; (ii) IDE depletion, in IDE-silenced cells, increase the efficacy of proteasome inhibition by the drug as suggested by the molecular studies herein reported. Thus, further experimental settings are required to fully clarify these possible scenarios.

Moreover, the results obtained in rMC1 cells (Figure 5), challenged with CFZ over a limited time-interval, showing a robust increase of IDE immunostaining in correspondence of the single-capped proteasome (i.e., 19S-20S complex, see Figure 5), indirectly confirm that the dynamics of interaction of IDE with proteasome assemblies can be mutually influenced by CFZ also in a living system.

As a CFZ-unrelated comment to these findings, since previous data on different human cell lines have reported that IDE preferentially binds the 20S and the single-capped 26S holoenzyme, at first sight, it might be surprising that we cannot detect IDE localization with the 20S in rMC1 cells. However, it must be pointed out that the distribution of proteasome assembly populations (i.e., 20S, 19S-20S, 19S-20S-19S) strongly differs among various cell lines and in different experimental conditions. Therefore, it is reasonable to expect proteasome interaction with IDE to differ in relation to the cell line investigated and to the specific metabolic conditions (see Figure 5B). Although the deepening of this aspect is outside the scope of this paper, it may be worth pointing out that CFZ administration turns out to decrease the immunodetection of proteasome particles by both anti-19S (e.g., Rpt5 subunit) and anti-20S (e.g., α7) antibodies (see Figure 5B,D), a finding recently observed also for BTZ when administered to some human cell lines [50].

However, no decrease of individual subunit (e.g., α7) can be actually detected under denaturing and reducing conditions, a finding which somewhat rules out that CFZ-bound proteasomes are degraded through autophagy; therefore, it would be relevant to disclose whether the catalytic inhibition, as that imposed by the drug, affects proteasome structural stability or the observation simply reflects a technical artefact, such as impaired epitope recognition by antibodies in the presence of accumulated and undigested substrates. Furthermore, the lack of variation of IDE immunostaining under denaturing and reducing conditions somehow excludes that IDE accumulation is due to its proteolytic processing by the single-capped particle. Over this short time interval, this would have been surprising, since it would have implied that IDE is a short-lived protein, a finding quite in contrast with the known biological features of the enzyme [14].

Although a more detailed description of this mechanism would require a deeper investigation, which is beyond the scope of this work, indeed we can assess that it is linked to the IDE-dependent structural and functional effect on 20S proteasome activity, previously described [13].

Due to its role in the clearance of insulin, the identification of drugs that selectively modulate IDE activity toward insulin is proposed by many authors as a promising therapeutic strategy for the cure of TDM2 and of its complications, such as diabetic neuropathy and retinopathy. However, the development of this therapeutic strategy should take into consideration also the new roles of IDE in the regulation of proteostasis, such as the modulation of proteasome activity; thus, it is undoubtable that IDE dysregulation contributes to proteostasis unbalance, even though its relevance in vivo remains still poorly characterized. Therefore, since IDE alteration seems to contribute to protein misfolding-related diseases and UPS dysfunctionality is a hallmark of the onset and progression of human pathologies, spanning from neurodegenerative and neurodevelopmental disorders to cancer [1,51,52,53], a deeper understanding of the biological significance of IDE–proteasome interaction and of the factors that can affect this interaction is mandatory.

## Data Availability

Original data will be available upon reasonable request to Dr. Grazia Tundo.

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
