# Peer review of "Insulin-Degrading Enzyme Is a Non Proteasomal Target of Carfilzomib and Affects the 20S Proteasome Inhibition by the Drug"

_biomolecules, 2022, doi:10.3390/biom12020315_

Round 1
Reviewer 1 Report
authors have satisfactorily addressed the reviewer's comments.
Author Response
We would thank the reviewer for your comments.
Reviewer 2 Report
The article by Coletta describes the finding that insulin-degrading enzyme (IDE) enhances the effects of carfilzomib by enhancing its proteasome binding and also targets to IDE. The binding to IDE is however significantly weaker than carfilzomib (CFZ) binding to the proteasome, thus the physiological relevance of this is not clear. Overall, this study is a follow-up on the earlier work on interaction of IDE and the proteasome. Some role of CFZ may be there, but very likely not at physiologically relevant conditions, which bring up an important point, which is whether this study provides new inside and whether one could actually state that IDE is a target of CFZ.
Figure 1. The activity is listed as a relative activity, normalized to 1. The actual reading should (Rfu) also be presented since it is not known whether or not the data is near background activities, since the 20S proteasome was not activated (with heat, SDS or by using open gate mutant).
The authors state that CFZ is a more powerful inhibitor of 20S proteasome in the presence of IDE. This should be validated by overexpressing (and/or knockdown) of IDE and comparison of the CC50 values of CFZ compared to controls. Otherwise, the significance of this in vitro systems may be considered irrelevant.
The amount of CFZ used in the experiments is also far above a clinical amount or its CC50. Are the effects observed also obtained at physiologically relevant concentrations?
The dose/time response in Fig. 2B appears inverted for the higher concentrations. This is odd and should be explained.
Figure 5. at micromolar CFZ concentrations, there is a ~10% effect on the peptide fragments. Again, the effects are very minimum at concentrations that are not physiologically relevant for this compound. This strongly questions the relevance of the findings as a whole.
For publications the authors need to conduct additional experiments that indicate an effect at physiologically relevant conditions.
Round 2
Reviewer 2 Report
The authors did not address the prior concerns by this reviewer.
The authors still normalize their data to 1, and not providing Rfu values to indicate the rates of degradation. It is therefore impossible to get a true sense of the activity of the proteasome. This reviewer is very much aware that this is a kinetic experiment and that the authors analyzed slopes.
The activities of the peptide degradation in the Mass Spec experiment still shows an extremely minor effect at concentrations that are irrelevant for any physiological significance.
In addition, as described previously, the CC50 of CFZ should be calculate (n=3) in their cell line of choice, with and without addition of the siRNA and the statistical significance between the two CC50 values should be determined to indicate whether or not IDE affects CFZ cytotoxicity. The authors chose to take single doses at single time-points, which is an unreliable method.
The physiological relevance of this work remains highly questionable as it is still not address in the revised manuscript
This manuscript is a resubmission of an earlier submission. The following is a list of the peer review reports and author responses from that submission.
Round 1
Reviewer 1 Report
Review of INSULIN-DEGRADING ENZYME IS A NON PROTEASOMAL TARGET OF CARFILZOMIB AND AFFECTS THE 20S PROTEASOME INHIBITION BY THE DRUG.
The article by Tundo GR and coworkers is describes novel observations regarding the association of IDE with proteasomes complexes and the inhibitory effect of carfilzomib on both 20S proteasomes as well as on IDE. Importantly, the interaction of the Insulin-Degrading Enzyme with 20S enhances the inhibitory effect of Carfilzomib. The article is clearly written and the results are described critically.
A few points could benefit from some additional clarifications:
- The final statement in the abstract is “the IDE-20S complex is an even better target of Carfilzomib that 20S alone “, yet in Figure 2D the authors find that (p.7) “robust IDE immunostaining in correspondence of the single capped proteasome, but not of free 20S…” and on p.15 “…we cannot detect IDE localization with the 20S in rMC1 cell lines…”. Please clarify and discuss which proteasome specie is the one that binds IDE and which is inhibited by CFZ?
- Equation 1 and kinetic model of CFZ inhibition. Please explain what is each parameter in the equation and why was this kinetic model chosen. For instance, is [CFZ] the concentration of CFZ of the CFZ-inhibited enzyme? In a scenario that δ(delta)=0 (a single binding site) then the equation appears to collapse to 1/[CFZ]^2. Is that accurately reflecting the residual enzymatic activity? This model assumes that the second site depends on the first site (K1 δ vs K1) could there be two independent sites (K1 and K2)?
Have the authors considered a Michaelis-Menten representation of enzymatic activity (rather than residual activity Fig 1) over substrate concentrations (fluorogenic peptide hydrolysis) repeated with increasing CFZ concentrations? What kind of inhibition would CFZ yield? Would there still be need for allostery between two sites? If the M&M analysis is not relevant or not necessary, please explain and rationalize the use of the model that depends on scheme 1 and equation 1.
- Question regarding scheme 1 and figure 1: The results are clear and the conclusion that “a simple competitive inhibitor is clearly unsatisfactory” for CFZ inhibition is warranted. Just out of curiosity, has asymmetry between the two beta5 protease site sin the 20S been raised or addressed in the literature? If not, then how do the authors explain that the two sites are not equivalent (given that the 20S barrel is supposedly symmetric)? Could binding of an inhibitor to one beta5 site allosterically affect the other beta5 active site? It would be nice if the authors discuss this a bit more in the discussion. Just to clarify, the kinetics in Figure 1 are with purified 20S is that correct? Has the purity of 20S been verified? Could there be some proteasome activators (19S, 11S, PA28, PA200) bound to some complexes, potentially inducing inherent asymmetry between two halves of the 20S? In Fig 2, for instance, IDE appears to preferentially bind to 19S-20S complexes rather than to free 20S complexes, could this somehow explain the results in Fig 1?
- Figure 2. The association of CFZ and/or IDE with “proteasome assemblies” was performed in cell extract according to the text, is that correct? Have the authors considered testing wt. purified 20S or 26S proteasomes (as the kinetics in Figure 1 were performed on isolated purified 20S).
Panel B vs panel D. In describing Fig 2B, the authors state that “proteasome immunostaining reduced in the presence of CFZ”. Nevertheless, IDE association appears to increase with CFZ incubation (Fig 2D). Could the authors clarify this point? If there is more 19S-20S complexes (right lanes in panel D) would they stain more with 19S or 20S antibodies (panel B)?
Author Response
Reviewer#1
Reviewer: A few points could benefit from some additional clarifications:
The final statement in the abstract is “the IDE-20S complex is an even better target of Carfilzomib that 20S alone “, yet in Figure 2D the authors find that (p.7) “robust IDE immunostaining in correspondence of the single capped proteasome, but not of free 20S…” and on p.15 “…we cannot detect IDE localization with the 20S in rMC1 cell lines…”. Please clarify and discuss which proteasome specie is the one that binds IDE and which is inhibited by CFZ?
Authors: In our previous paper (Sbardella et al., 2018) we have reported that IDE co-localizes preferentially with the 20S proteasome and with the single capped particle (19S-20S). In rMC1 cells, unlike other cell lines tested so far, IDE immunostaining is detected in correspondence of the single capped proteasome, but not of free 20S. At first sight, this result could appear surprising. Anyway, we would underline as the distribution of different proteasome assembly populations (i.e., 20S, 19S-20S, 19S-20S-19S) differs among various cell lines and in different experimental conditions. Thus, also the interaction with IDE is reasonable to differ in relation to the cell line investigated and to the dynamics of cell metabolism. We would also recall that CFZ inhibits the catalytic activity of the 20S, regardless of the presence of regulatory particles. Therefore, the inhibitory effect, as it comes out from in vitro analysis, in any case, is due to its interaction with the 20S proteasome.
Reviewer: Equation 1 and kinetic model of CFZ inhibition. Please explain what is each parameter in the equation and why was this kinetic model chosen. For instance, is [CFZ] the concentration of CFZ of the CFZ-inhibited enzyme? In a scenario that δ(delta)=0 (a single binding site) then the equation appears to collapse to 1/[CFZ]^2. Is that accurately reflecting the residual enzymatic activity? This model assumes that the second site depends on the first site (K1 δ vs K1) could there be two independent sites (K1 and K2)?
Authors: We would like to underline that, according to experimental data, the reaction of CFZ with 20S CP is described with the simplest possible Scheme, which accounts for the fact that in 20S indeed there are two chymotrypsin-like binding sites for CFZ. Therefore, Eq. (1) does not take into consideration the possibility of only one binding site (in such a case the equation would be different because the Scheme is different). The case δ = 0 does not refer to a single binding site, but instead to the eventuality that binding of CFZ to one site impairs completely CFZ binding to the second one, rendering impossible the complete abolishment of the enzymatic activity. A more realistic possibility is that the binding of CFZ to one site reduces the affinity of CFZ for the second site; this occurrence corresponds to δ > 1 (negative cooperativity), which is analogous to K1 < K2, as asked by the reviewer. Thanks to the reviewer observation, we have included in the revised version the explanation of the meaning of parameters of Scheme 1 and Eq. (1), as well as a more explicit interpretation of the results.
Reviewer: Have the authors considered a Michaelis-Menten representation of enzymatic activity (rather than residual activity Fig 1) over substrate concentrations (fluorogenic peptide hydrolysis) repeated with increasing CFZ concentrations? What kind of inhibition would CFZ yield? Would there still be need for allostery between two sites? If the M&M analysis is not relevant or not necessary, please explain and rationalize the use of the model that depends on scheme 1 and equation 1.
Authors: We have carried out data at different substrate concentrations and we could draw Lineweaver-Burk plots, but, in our opinion, the visual message would be much more complicated, and the immediate comparison with the effect of IDE on the inhibitory process would be lost. It is known that CFZ is a competitive inhibitor of proteasome. In this paper, we want only to show the effect of IDE on the CFZ inhibition toward proteasome, and this effect is better displayed by the plot reported in Fig. 1.
Reviewer: Question regarding scheme 1 and figure 1: The results are clear and the conclusion that “a simple competitive inhibitor is clearly unsatisfactory” for CFZ inhibition is warranted. Just out of curiosity, has asymmetry between the two beta5 protease site sin the 20S been raised or addressed in the literature? If not, then how do the authors explain that the two sites are not equivalent (given that the 20S barrel is supposedly symmetric)? Could binding of an inhibitor to one beta5 site allosterically affect the other beta5 active site? It would be nice if the authors discuss this a bit more in the discussion. Just to clarify, the kinetics in Figure 1 are with purified 20S is that correct? Has the purity of 20S been verified? Could there be some proteasome activators (19S, 11S, PA28, PA200) bound to some complexes, potentially inducing inherent asymmetry between two halves of the 20S? In Fig 2, for instance, IDE appears to preferentially bind to 19S-20S complexes rather than to free 20S complexes, could this somehow explain the results in Fig 1?
Authors: The point raised by the reviewer is correct, and we explicitly claim that the two binding sites of the β5 subunits are functionally related, such that there is an allosteric interaction between the two sites (expressed by the parameter δ, discussed above). In any case, as suggested by the reviewer, in the revised version we have outlined this point more extensively. As far as the purity of the 20S, it was assayed and, in the revised version of manuscript, a typical SDS-PAGE pattern of 20S subunits under denaturing and reducing condition has been included as Supplementary Figure 1, wherefore any contamination by 19S subunits has been identified in the 20S preparation used for the kinetic experiments.
Reviewer: Figure 2. The association of CFZ and/or IDE with “proteasome assemblies” was performed in cell extract according to the text, is that correct? Have the authors considered testing wt. purified 20S or 26S proteasomes (as the kinetics in Figure 1 were performed on isolated purified 20S).
Panel B vs panel D. In describing Fig 2B, the authors state that “proteasome immunostaining reduced in the presence of CFZ”. Nevertheless, IDE association appears to increase with CFZ incubation (Fig 2D). Could the authors clarify this point? If there is more 19S-20S complexes (right lanes in panel D) would they stain more with 19S or 20S antibodies (panel B)?
Authors: Yes, it is correct. In this cell model, the association of CFZ and/or IDE with proteasome assemblies was analyzed in crude cell extracts by native gel electrophoresis. The kinetic analysis was instead performed by using a commercial 20S CP, whose purity was previously assayed by SDS-PAGE (see Supplementary Figure 1 in the revised manuscript). The hypothesis to test the effect directly on 20S preparation purified by this cell is very intriguing and it could be taken into consideration for additional investigations. However, being the first work in which the effect of IDE on CFZ-20S complex is investigated, we planned to test the kinetic analysis of this phenomena on commercial purified enzymes which have been previously used for biochemical investigations by our group.
Concerning the possibility of using the isolated 26S proteasome, this is a very relevant point and we firmly believe that it must be carefully taken into consideration for future studies; at the moment, since we previously reported that IDE may act as a competitor of the 19S (Sbardella et al., 2018), we are limited by the fact that IDE administration to the isolated and purified 26S quickly triggers 20S and 19S dissociation (forming IDE:20S complexes).
Reviewer 2 Report
This manuscript suffers from several major flaws and raises numerous major questions:
- The main conclusion that insulin-degrading enzyme is a non-proteasomal target of carfilzomib is not supported by data; Fig. 5 clearly shows that Cfz does not inhibit IDE
- The purity of commercial IDE preparation is not clear; what if effects described on Fig. 1 and 2 were caused by contamination?
- Experiments on Fig.1 used 20-fold molar excess of IDE2 over the proteasome; what is IDE2:proteasome ratio in cells? If it is lower then in this experiment, the result may be an in vitro artefact;
- The specificity of IDE2 antibody has not been charachterized;
- Even if IDE2 band of Fig. 2D is indeed IDE2 bands, apparent co-migration of IDE2 with proteasome on the native gel does not mean that they are associated together; what if proteasome inhibition cause ID2 aggregation or a complex formation of IDE2 with different protein; immunoprecipitation is needed to prove association
- If authors are trying to make the claim that IDE affects proteasome inhibition by carfilzomib, and not by other proteasome inhibitors, where is the data that it does not affect inhibition by other inhibitors (e.g., bortezomib)?
- What will be the results of experiment on Fig. 2B if IDE2 is knocked down by siRNA or knocked out by CRISPR? Utilizing active site mutant would also help
Minor points:
- The statement that non-proteasomal targets of Cfz has not been reported yet is not correct, please see PMID: 27503896
- English is bad
- The manuscript contains many typos and leaves an impression that it has not been proofread before submission
Author Response
Dear Editor,
here attached a revised version of the manuscript “Biomolecules-1152527” “INSULIN-DEGRADING ENZYME IS A NON PROTEASOMAL TARGET OF CARFILZOMIB AND AFFECTS THE 20S PROTEASOME INHIBITION BY THE DRUG” by G. R. Tundo, D. Sbardella, F. Oddone, G. Grasso, S. Marini, A. M. Santoro, D. Milardi, F. Bellia, G. Macari, F, Polticelli, P. Cascio, M. C. Parravano, and myself.
First of all, we would thank all reviewers for accurate criticisms, which indeed helped us in improving the manuscript. Hereafter, is a point-by-point reply to their criticisms.
All modifications in the revised version are highlighted in yellow.
We hope that the revised manuscript may be appealing for the Biomolecules readers.
Yours Sincerely
Prof. Massimo COLETTA
Reviewer #2
Reviewer: The main conclusion that insulin-degrading enzyme is a non-proteasomal target of carfilzomib is not supported by data; Fig. 5 clearly shows that Cfz does not inhibit IDE
Authors: Actually, Fig. 4 demonstrates unequivocally that CFZ inhibits IDE, even though over a concentration range 10-fold higher than for 20S proteasome. In any case, it is undoubtable that CFZ inhibits IDE at sub-micromolar concentrations (see also Table 3), which is a range over which most specific inhibitors do not inhibit an enzyme. Concerning data on insulin degradation (i.e., Fig. 5), we have modified it, normalizing data to the Total Ion Current (TIC) and adding statistical significance. It is now easy to estimate the inhibiting role of CFZ on IDE activity towards insulin. It is clear that the inhibiting effect is very small (no more than 10%) and the results, which are somewhat different from the ones obtained in the case of the fluorogenic substrate used for the activity assay, can be explained by considering the different experimental conditions requested by the MS assay (the concentration of IDE is 10-fold higher and therefore the oligomeric distribution of the enzyme is different). In any case, since our interpretation of the CFZ inhibitory mechanism envisages the possibility that CFZ is primarily a non-competitive inhibitor of IDE (as suggested by Fig. 4), it is not completely unexpected that the direct inhibitory effect on a substrate is less evident. Thanks to the reviewer criticism, we have more deeply re-analyzed the data and in the revised version we have extended the discussion on this point.
Reviewer: The purity of commercial IDE preparation is not clear; what if effects described on Fig. 1 and 2 were caused by contamination?
Authors: In the revised version of the manuscript, the SDS-PAGE of recombinant IDE that was used for all experiments has been included. We would like to underline that the purity of IDE preparation was tested before performing the experiments reported.
Reviewer: Experiments on Fig.1 used 20-fold molar excess of IDE2 over the proteasome; what is IDE2:proteasome ratio in cells? If it is lower then in this experiment, the result may be an in vitro artefact;
Authors: We would like to underline that the choice of IDE concentrations was not arbitrary, but it was performed based on a previous publication (Sbardella et al., 2018) in which the authors have characterized through a biochemical and computational approach the effect of IDE on 20S activity. Based on these results, it has been chosen the concentration of IDE that induces the maximum inhibitory effect on 20S activity. Concerning IDE-proteasome ratio, it is plausible that it depends on the specific cellular model investigated and by metabolic conditions of the cell. It is undoubtable that to unveil the biological significance of IDE-20S interaction and the exact role of IDE in the modulation of 20S activity several studies should be carried out and that the determination of IDE-proteasome ratio is a crucial point, but it goes beyond the purpose of the present paper. With respect to this, if we consider IDE a Proteasome Interacting Protein, it is worth saying that it was found to modulate the proteasome at a very low concentration (i.e., nanomolar range), several orders of magnitude lower than that reported by other authors for other PIPs (see Moskovitz et al.,2015).
Reviewer: The specificity of IDE2 antibody has not been charachterized;
Authors: Indeed this is a right point, but we actually verified the specificity of the antibody in the same cell line, herein used, upon silencing IDE expression through delivery of an anti-sense oligonucleotide (and a non-targeting pool as internal control). However, we must say that the finding shown below is part of a paper on a different topic which will be matter of a forthcoming publication. Furthermore, IDE specificity was further tested against rIDE by Wester blotting, see Supplementary Figure 1.
See uploaded file for figure
Figure: rMC1 were grown in Low Glucose DMEM and stimulated with a small interfering oligonucleotide (siRNA) targeting IDE or with a non-targeting pool of oligonucleotides (Scrambled) for 72h. As internal control, cells were further treated with siRNA pools vehicle (Ctrl). After stimulation, cells were harvested, lysed in RIPA and analysed by denaturing and reducing Western blotting. IDE was probed with the antibody used throughout this study. A representative experiment is shown. One-way ANOVA followed by Tukey’s post-hoc significance test (p<0.0001).
Reviewer: Even if IDE2 band of Fig. 2D is indeed IDE2 bands, apparent co-migration of IDE2 with proteasome on the native gel does not mean that they are associated together; what if proteasome inhibition cause ID2 aggregation or a complex formation of IDE2 with different protein; immunoprecipitation is needed to prove association
Authors: We acknowledge this criticism which actually helps us to better clarify this point. The assumption, made in the original discussion of the manuscript, was drawn on the basis of a bulk of independent evidences on IDE-proteasome interaction in almost every cell line investigated by us and by other authors. Specifically, IDE was found to interact with this proteasome assembly by glycerol-gradient purification (Duckworth WC, Bennett RG, Hamel FG. Insulin acts intracellularly on proteasomes through insulin-degrading enzyme. Biochem Biophys Res Commun. 1998 Mar 17;244(2):390-4. doi: 10.1006/bbrc.1998.8276. PMID: 9514933; Bennett RG, Hamel FG, Duckworth WC. Characterization of the insulin inhibition of the peptidolytic activities of the insulin-degrading enzyme-proteasome complex. Diabetes. 1997 Feb;46(2):197-203. doi: 10.2337/diab.46.2.197. Erratum in: Diabetes 1997 Sep;46(9):1532. PMID: 9000694.), by native-gel electrophoresis coupled with Western blotting (as done here), by native Mass spectrometry (Sbardella et al., 2018, cited throughout the text), and, further, by co-localization and co-immunoprecipitation studies (Sbardella et al., 2015, cited throughout the text). We recognize that the observation done in this cell model is based only on one approach (Native-gel), but the lack of any evidence of IDE aggregation and non-specific electrophoretic migration was somehow supported by decades of research on this enzyme. However, we have added a further comment in the discussion which opens up to future investigations on additional scientific possibilities.
Reviewer: If authors are trying to make the claim that IDE affects proteasome inhibition by carfilzomib, and not by other proteasome inhibitors, where is the data that it does not affect inhibition by other inhibitors (e.g., bortezomib)?
Authors: No, we are not making this point. We do not have any information about IDE effect on proteasome inhibition by other proteasome inhibitors. We are simply saying that:
- CFZ cannot be considered a highly specific inhibitor of proteasome, since IDE is inhibited as well by CFZ with inhibitory constants which are only 10-fold higher than 20S;
- IDE binding to 20S proteasome enhances the inhibitory strength of CFZ on 20S proteasome
Reviewer: What will be the results of experiment on Fig. 2B if IDE2 is knocked down by siRNA or knocked out by CRISPR? Utilizing active site mutant would also help
Authors: The suggestions of reviewer are very intriguing. However, in our opinion, these points are very complex (especially the knocking out by CRISPR) and deserve many additional experiments that it is hard to perform during the short time available for resubmission.
We perfectly agree with the reviewer that the use of an active site mutant could help to unveil the basis of IDE –proteasome interaction and the authors are planning to study this interaction using R767 IDE mutant which lacks the ability to dimerize. However, again, in our opinion, the experiments proposed go beyond the original purpose of the present paper.
Reviewer: Minor points:
- The statement that non-proteasomal targets of Cfz has not been reported yet is not correct, please see PMID: 27503896
- English is bad
- The manuscript contains many typos and leaves an impression that it has not been proofread before submission
Authors: We thank the reviewer for having raised our attention to an additional paper, which is along our line.
According to reviewer suggestions, the authors have performed a revision of the manuscript.

Reviewer 3 Report
The authors show that Insulin-Degrading Enzyme may be a target for the last generation proteasome inhibitor carfilzomib.
And elucidate kinetic parameters of IDE inhibition by carfilzomib. Though the results of the work are potentially interesting
their presentation and interpretation should be carefully reconsidered.
Comments:
Proteasome substrates are not "early-lived" proteins but short-lived proteins. IDE was used at 20 nM concentration; it is unclear what concentration is used for 20S proteasome.
The authors need to describe in detail the calculation of peptide concentrations measured by LC-MS. Did they use spike-in standards to make measurements more accurate?
Figure 4. No results on the statistical significance of the observed differences have been reported.
Results suggesting that CFZ inhibits insulin degradation by the IDE (Figure 5) are not convincing. No results on the statistical significance of the observed differences have been reported.
The observed slight differences could be the consequence of the increased DMSO concentration or other CFZ solvent used for the experiment. Then, the proteasome inhibitors are not used at concentrations that could potentially affect IDE activity on insulin, and CFZ is not a selective IDE inhibitor. These last points ruin the basis for discussion of selective IDE inhibitors and their significance in TDM2 therapy.
Author Response
Dear Editor,
here attached a revised version of the manuscript “Biomolecules-1152527” “INSULIN-DEGRADING ENZYME IS A NON PROTEASOMAL TARGET OF CARFILZOMIB AND AFFECTS THE 20S PROTEASOME INHIBITION BY THE DRUG” by G. R. Tundo, D. Sbardella, F. Oddone, G. Grasso, S. Marini, A. M. Santoro, D. Milardi, F. Bellia, G. Macari, F, Polticelli, P. Cascio, M. C. Parravano, and myself.
First of all, we would thank all reviewers for accurate criticisms, which indeed helped us in improving the manuscript. Hereafter, is a point-by-point reply to their criticisms.
All modifications in the revised version are highlighted in yellow.
We hope that the revised manuscript may be appealing for the Biomolecules readers.
Yours Sincerely
Prof. Massimo COLETTA
Reviewer #3
Reviewer: Proteasome substrates are not "early-lived" proteins but short-lived proteins. IDE was used at 20 nM concentration; it is unclear what concentration is used for 20S proteasome.
The authors need to describe in detail the calculation of peptide concentrations measured by LC-MS. Did they use spike-in standards to make measurements more accurate?
Authors: The criticism of the reviewer is correct. We have included “short lived” in the place of “early-lived” proteins. The concentration of 20S proteasome used is 1 nM, as now indicated in “Materials and Method” section.
Although we have experimentally checked the very good intra-day and inter-day reproducibility of the absolute ion intensity for each insulin peptide peak detected by MS, we have normalized the data to the Total Ion Current (TIC) in order to make the measurements more accurate. The new Figure 5 reports now the relative areas of the peaks as well as the statistical significance of measurements.
Reviewer: Figure 4. No results on the statistical significance of the observed differences have been reported.
Authors: We acknowledge this point, even though the error bars are reflecting the statistical significance of experimental points in Fig. 4. However, in the revised version of manuscript we have pointed out the statistical significance of observed difference for Fig. 4.
Reviewer: Results suggesting that CFZ inhibits insulin degradation by the IDE (Figure 5) are not convincing. No results on the statistical significance of the observed differences have been reported.
Authors: The criticism of reviewer is correct. Statistical significance of the MS results have been added in the new Figure 5 (see above). Discussion has also been improved in the text.
Reviewer: The observed slight differences could be the consequence of the increased DMSO concentration or other CFZ solvent used for the experiment. Then, the proteasome inhibitors are not used at concentrations that could potentially affect IDE activity on insulin, and CFZ is not a selective IDE inhibitor. These last points ruin the basis for discussion of selective IDE inhibitors and their significance in TDM2 therapy.
Authors: As pointed out in the previous comment, we have considered other explanations for the observed results, significantly modifying the revised text.
Round 2
Reviewer 3 Report
The authors addressed the comments and improved the manuscript. So, I it could be recommended for publication in the Journal.